# Comparative Efficacy of Tocilizumab and Baricitinib Administration in COVID-19 Treatment: A Retrospective Cohort Study

**DOI:** 10.3390/medicina58040513

**Published:** 2022-04-04

**Authors:** Yuichi Kojima, Sho Nakakubo, Nozomu Takei, Keisuke Kamada, Yu Yamashita, Junichi Nakamura, Munehiro Matsumoto, Hiroshi Horii, Kazuki Sato, Hideki Shima, Masaru Suzuki, Satoshi Konno

**Affiliations:** 1Department of Respiratory Medicine, Faculty of Medicine and Graduate School of Medicine, Hokkaido University, North 15 West 7, Kita-ku, Sapporo 060-8638, Japan; yu1.kojima@pop.med.hokudai.ac.jp (Y.K.); ntakei@pop.med.hokudai.ac.jp (N.T.); keisukekmd@gmail.com (K.K.); junnaka.3n3.11@gmail.com (Y.Y.); mune@pop.med.hokudai.ac.jp (J.N.); h.holy724@gmail.com (M.M.); sakazuki@pop.med.hokudai.ac.jp (H.H.); hideki.shima@pop.med.hokudai.ac.jp (K.S.); suzumasa@med.hokudai.ac.jp (H.S.); satkonno@med.hokudai.ac.jp (M.S.); yuyama@pop.med.hokudai.ac.jp (S.K.); 2Department of Mycobacterium Reference and Research, The Research Institute of Tuberculosis, Japan Anti-Tuberculosis Association, 3-1-24, Matsuyama Kiyose, Tokyo 101-0061, Japan; 3Department of Epidemiology and Clinical Research, The Research Institute of Tuberculosis, Japan Anti-Tuberculosis Association, 3-1-24, Matsuyama Kiyose, Tokyo 101-0061, Japan; 4Department of Respiratory Medicine 1, Obihiro Kosei General Hospital, Obihiro 080-0016, Japan

**Keywords:** COVID-19, tocilizumab, baricitinib, retrospective study

## Abstract

*Background and Objectives:* Tocilizumab and baricitinib have been observed to improve the outcomes of patients with coronavirus disease 2019 (COVID-19). However, a comparative evaluation of these drugs has not been performed. *Materials and Methods:* A retrospective, single-center study was conducted using the data of COVID-19 patients admitted to Hokkaido University hospital between April 2020 and September 2021, who were treated with tocilizumab or baricitinib. The clinical characteristics of the patients who received tocilizumab were compared to those of patients who received baricitinib. Univariate and multivariate logistic regression analyses of the outcomes of all-cause mortality and improvement in respiratory status were performed. The development of secondary infection events was analyzed using the Kaplan–Meier method and the log-rank test. *Results:* Of the 459 patients hospitalized with COVID-19 during the study, 64 received tocilizumab treatment and 34 baricitinib treatment, and those 98 patients were included in the study. Most patients were treated with concomitant steroids and exhibited the same severity level at the initiation of drug treatment. When compared to each other, neither tocilizumab nor baricitinib use were associated with all-cause mortality or improvement in respiratory status within 28 days from drug administration. *Conclusions:* Age, chronic renal disease and early administration of TCZ or BRT from the onset of COVID-19 were independent prognostic factors for all-cause mortality, whereas anti-viral drug use and the severity of COVID-19 at baseline were associated with an improvement in respiratory status. Secondary infection-free survival rates of patients treated with tocilizumab and those treated with baricitinib did not significantly differ. The results suggest that both tocilizumab and baricitinib could be clinically equivalent agents of choice in treatment of COVID-19.

## 1. Introduction

Coronavirus disease 2019 (COVID-19), caused by severe acute respiratory syndrome coronavirus 2 (SARS-CoV-2), continues to spread worldwide. As various treatment methods have been established and vaccination has progressed, the COVID-19 situation has entered a new phase. However, research to determine the optimal treatment for severe COVID-19 is ongoing.

Corticosteroids are potent cytokine inhibitors that act through several mechanisms, but primarily by inhibiting the NF-κB transcription factor [1]. Steroid administration was the first effective treatment identified for severe COVID-19, and is now regarded as standard treatment [2,3]. Subsequently, several studies were conducted to determine whether the addition of immunomodulators to the standard treatment could improve prognosis. Tocilizumab (TCZ) is an anti-interleukin-6 receptor-alpha monoclonal antibody [4]. The immunological effects of TCZ are expected to include induction/expansion of B regulatory cells, and a decreased expression of inflammatory cytokine and chemokine genes [5]. REMAP-CAP and RECOVERY trials were used to evaluate the add-on effect of TCZ to the standard care of hospitalized patients with severe-to-critical COVID-19 and showed that TCZ reduces mortality or prolongs organ support-free days [6,7]. Baricitinib (BRT) is a Janus kinase (JAK) inhibitor with high selectivity for JAK1 and JAK2 molecules of the JAK family [8]. The JAK/STAT pathway is located downstream of several cytokines that are increased in HLH and is, therefore, a target for inhibiting signaling of multiple cytokine pathways [9]. When combined with remdesivir in the treatment of severe COVID-19 in the ACTT-2 trial, BRT was initially observed to reduce the recovery duration [10]. Furthermore, in the COV-BARRIER trial, treatment with BRT, in addition to standard care, was associated with reduced mortality among adults hospitalized with COVID-19 [11]. In these trials, both TCZ and BRT were found to be particularly effective in reducing mortality among patients with a high demand for oxygen.

Based on the evidence, the current guidelines recommend that TCZ and BRT be administered along with steroids to patients with severe COVID-19 requiring high-flow oxygen and non-invasive mechanical ventilation, and those with rapidly increasing oxygen needs and systemic inflammation [12,13]. However, to our knowledge, no comparative study has verified the superiority of the efficacy of TCZ or BRT against COVID-19; therefore, the current international recommendation does not accord precedence to either treatment [12]. This study aimed to perform a comparative evaluation of TCZ and BRT in the treatment of patients with COVID-19.

## 2. Materials and Methods

### 2.1. Patient Consent Statement

This single-center, retrospective cohort study was approved by Hokkaido University Hospital Division of Clinical Research Administration (Research No. 020-0107). The analyses were conducted retrospectively using existing samples collected in the course of routine clinical practice, with no additional risks to the patients. Therefore, the requirement for obtaining informed consent from individual participants was waived by the aforementioned ethics committee. Patient information was anonymized to the extent possible.

### 2.2. Patients

The study participants comprised patients with COVID-19 who were admitted to the Hokkaido University Hospital between April 2020 and September 2021. All of the patients were confirmed to be positive for SARS-CoV-2 through polymerase chain reaction (PCR). Patients treated with TCZ or BRT for COVID-19 were selected for the present analysis. Cases where both drugs were administered during the course of treatment were disregarded.

### 2.3. Data Collection

The clinical data (age, sex, body mass index, history of smoking, history of vaccination, comorbidities, respiratory status and severity, days from the onset of COVID-19, treatment protocol laboratory data, and clinical outcome) were collected from medical records. We categorized the severity of COVID-19 as follows; severity level 1: hospitalized but not requiring supplemental oxygen; severity level 2: hospitalized and requiring supplemental oxygen ≤4 L per minute (L/min); severity level 3: hospitalized and requiring oxygen therapy ≥5 L/min, including receiving nasal high-flow oxygen therapy, non-rebreather, or noninvasive mechanical ventilation; severity level 4: receiving invasive mechanical ventilation upon administration. This severity classification was created because the criterion for administering TCZ or BRT in our hospital was a deterioration of ≥5 L/min in the oxygen administration rate.

We set the following clinical endpoints: all-cause mortality, improvement in respiratory status, and development of secondary infection events within 28 days from administration of TCZ or BRT. Improvement in respiratory status after the initiation of TCZ or BRT treatment was regarded as recovery in severity levels 1 and 2. Patients with a baseline severity of 1 or 2 were treated as “improved” if they remained at severity 1 or 2 at 28 days. Secondary infection events included pneumonia, bacteremia, urinary tract infection, and fungal infection, which required antibiotic treatment.

### 2.4. Statistical Analysis

Continuous data are expressed as median and interquartile range (IQR). Categorical data are expressed as absolute numbers and percentages. The Wilcoxon rank sum test or Kruskal–Wallis test was used to compare differences in continuous variables, and the chi-square test or Fisher’s exact test was used to evaluate differences between categorical variables. Clinical outcomes, including all-cause mortality, improvement in respiratory status, and development of secondary infection events, were analyzed using univariable and multivariate logistic regression models, with the odds ratio (OR) and 95% confidence intervals (CI). Variables with *p* < 0.1 in the univariate analysis were inputted into the multivariate models. Infection-free survival was evaluated using the Kaplan–Meier method and a log-rank test. As a sensitivity analysis, propensity scores were calculated by logistic regression and used as matching parameters to adjust for measured confounders. Patients treated with tocilizumab were matched 1:1 to patients with baricitinib using nearest neighbor matching with a caliper of 0.2. All of the *p*-values were 2-tailed, with statistical significance set at *p* < 0.05. JMP (SAS Institute Inc., Cary, NC, USA) was used for all of the statistical processing.

## 3. Results

### 3.1. Study Population

A total of 459 patients diagnosed with COVID-19 were admitted to the Hokkaido University Hospital during the study period. To treat the symptoms of COVID-19, 100 of these patients received either TCZ or BRT, 64 were treated with TCZ (TCZ group), and 34 were treated with BRT (BRT group). Two patients initially received BRT treatment but later received TCZ (Figure 1), and were excluded from the study. Subsequent analyses were performed on the remaining 98 individuals.

### 3.2. Baseline Characteristics

The median age of the total number of patients (*n* = 98) was 60.5 years, and 74.5% of them were males (Table 1). Compared with the TCZ group (*n* = 64), the BRT group (*n* = 34) had a low age range (58.5 vs. 65.5 years, *p* = 0.03) and low prevalence of chronic heart disease (5.9% vs. 23.4%, *p* = 0.03). There were no significant differences in sex, smoking history, immunosuppressive drug use, obesity, chronic kidney disease, diabetes mellitus, collagen disease, hypertension, and comorbid respiratory disease. Only one patient—in the BRT group—was fully vaccinated with two doses of the vaccine against SARS-CoV-2. The number of days from the onset of illness to the administration of the drugs for the two groups was not significantly different (10 vs. 9 days, *p* = 0.50). Analysis of blood samples revealed that, compared to the BRT group, the TCZ group had a significantly lower eosinophil count and hemoglobin (0 vs. 0, *p* = 0.047, 13.8 vs. 14.5, *p* = 0.04, respectively), and high levels of lactate dehydrogenase (LDH), krebs von den Lungen-6 (KL-6), and D-dimer (540 vs. 470, *p* = 0.02, 444 vs. 319, *p* = 0.03, 1.5 vs. 1.0, *p* < 0.01, respectively).

All of the patients in the TCZ group received the drug as a single intravenous infusion. The dose of tocilizumab was established by bodyweight (8 mg/kg). Baricitinib was administered at a dose of 4 mg/day in the BRT group. Moreover, 2 mg/day was administered to patients who had a baseline eGFR of 30 to less than 60 mL/min/1.73 m^2^. Baricitinib was administered orally (or crushed for nasogastric tube administration) daily for up to 14 days or until the clinicians determined that the patient’s symptoms had improved. Most patients in both groups received steroid treatment for COVID-19 (98.4% in the TCZ and 97.1% in the BRT group). Heparin was administered more frequently and antivirals less frequently to the TCZ than to the BRT group (86.0% vs. 67.7%, *p* = 0.03, 70.3% vs. 88.2%, *p* = 0.046, respectively). Only one patient —in the BRT group—was treated with a combination of monoclonal antibodies (casirivimab and imdevimab). The severity of COVID-19 was similar in both groups at the time of treatment initiation, with 93.2% of the patients in TCZ group and 85.3% in BRT group having a severity level of ≥3.

A total of 14 patients died within 28 days of receiving the biological agents (TCZ group; 13, 20.3%, BRT group; 1, 2.94%). Respiratory status improved within 28 days of biologic therapy in 73 patients (74.5%), 43 (67.2%) in the TCZ group and 33 (88.2%) in the BRT group. 

BMI, body mass index; CRP, C-reactive protein; KL-6, Krebs von den Lungen-6; LDH, lactate dehydrogenase.

### 3.3. Risk Factors for Death within 28 Days from Initiating TCZ or BRT Treatment 

Among the group of patients treated with either TCZ or BRT (*n* = 98), univariate analysis showed that the use of TCZ was significantly associated with increased all-cause mortality when compared to the use of BRT. Additionally, advanced age, the presence of chronic kidney disease, administration of biological agents in less than seven days from the onset of COVID-19, and the lack of antiviral drug use were significantly associated with all-cause mortality (Table 2). In the multivariate analysis, relatively old age [OR = 1.10, 95% confidence interval (CI) 1.00–1.21, *p* = 0.02], presence of chronic kidney disease (OR = 43.10, 95% CI 2.71–686.04, *p* = 0.008), and early administration of TCZ or BRT from the onset of COVID-19 (OR = 18.09, 95% CI 1.70–192.47, *p* = 0.02) were observed to be independent risk factors for all-cause mortality within 28 days from treatment initiation. In contrast, the use of TCZ was not an independent prognostic factor for death (OR = 13.28, 95% CI 0.45–392.92, *p* = 0.13) (Table 2).

### 3.4. Factors Contributing to Improvement in Respiratory Status

The univariate logistic regression analysis showed that treatment with BRT, young age, absence of chronic heart disease, chronic kidney disease or hypertension, more than seven days from the onset to drug administration, and the use of any anti-viral drug significantly contributed to improvement in respiratory status (Table 3). However, based on the multivariate analysis, BRTtreatment did not result in an improved respiratory status as compared to TCZ (OR = 1.75, 95% CI 0.35–8.67, *p* = 0.50), whereas the use of the anti-viral drug was an independent contributing factor (OR = 6.5, 95% CI 1.13–37.56, *p* = 0.04). Early administration of TCZ or BRT was a risk factor that reduced the likelihood of improving respiratory status (OR = 0.82, 95% CI 0.02–0.40, *p* = 0.002).

### 3.5. Development of Secondary Infections

The rates of acquiring any secondary infection among patients within 28 days from initiation of treatment with TCZ and BRT were 15.6 and 14.7%, respectively. Univariate analysis did not identify any factors associated with the development of secondary infections after initiation of treatment (Appendix A). There was also no significant difference in the infection-free survival (*p* = 0.95) (Figure 2).

### 3.6. Sensitivity Analyses

Since the number of death events was low (14 out of 98 patients), a sensitivity analysis was performed to strengthen the results of this study. Age, history of chronic kidney disease, any respiratory disease, time from onset to administration ≤7 day, and any anti-viral drug were matched as confounders. The population after the propensity score matching consisted of 25 tocilizumab-users and 25 baricitinib-users. There was no significant difference in mortality, recovery rate of respiratory status, and secondary infection rate between the TCZ and BRT groups (8.0% vs. 4.0%, *p* = 0.55, 84.0% vs. 88.0%, *p* = 0.68, 12.0% vs. 16%, *p* = 0.68, respectively) (Appendix A).

## 4. Discussion

In this retrospective study, we compared the clinical characteristics between two groups of patients treated with TCZ and BRT for COVID-19. Multivariate analysis revealed that neither drug increased the risk of death within 28 days from treatment initiation when compared to each other. Age, underlying diseases, and early administration of the drugs were independent risk factors for all-cause mortality. Neither TCZ nor BRT showed a greater contribution to an improved respiratory status within 28 days from treatment initiation. The use of anti-viral drugs and late administration of TCZ or BRT significantly contributed to improving the respiratory status. No significant difference was observed in the development of secondary infections within 28 days from TCZ or BRT administration. Based on the results of our study, we expect that neither drug would be superior to the other in the treatment of COVID-19.

The results of this study showed that most of the patients who were treated with TCZ or BRT simultaneously received steroid therapy. In addition, there was no significant difference between the severities of COVID-19 in the two groups at the time of treatment initiation with TCZ or BRT. In our hospital, patients with COVID-19 who require oxygen are usually treated with steroids, and other immunomodulators are additionally administered to patients with increased oxygen demand, based on the treatment guidelines [12,13]. Although the present study was retrospective, the study was robust in that the baseline treatment and respiratory status of both groups were consistent. 

According to the results of the univariate analysis, TCZ seemed to increase the risk of mortality within 28 days from treatment initiation without improving the respiratory status when compared to BRT. However, it was not identified as a significant risk factor in the multivariate analysis, perhaps owing to the existence of multiple confounding factors with regards to the use of TCZ. Comparison between clinical characteristics of the TCZ and BRT groups revealed an older median age and a higher proportion of patients with chronic diseases in the TCZ group than in the BRT groups. Moreover, blood test results showed higher levels of LDH and D-dimer in the TCZ group than in the BRT group. Advanced age and underlying diseases are regarded as poor prognostic factors for COVID-19 [14,15]. Relatively high levels of LDH and D-dimer are associated with an increased mortality in COVID-19 and are known to be predictors of severe disease [16]. Although we did not find a significant difference in the severity of COVID-19 based on respiratory status, the TCZ group might have potentially been at a higher risk for a critical course. TCZ was shown to be effective earlier than BRT in the COVID-19 pandemic and was used earlier in clinical practice. In contrast, BRT was widely used in Japan upon approval for the treatment of COVID-19. In addition, the backgrounds of the patients admitted to our hospital differed depending on the timing of the spread of COVID-19. These circumstances might contribute to the bias in the clinical characteristics of the two groups.

Biological drugs are a risk factor for severe infection in rheumatoid arthritis [17]. Although the COVID-19 trials showed no difference in the incidence of infections in either TCZ or BRT groups, compared to that in the placebo [6,11], one retrospective study showed that the concomitant use of TCZ and methylprednisolone is a risk factor for bacteremia [18]. Furthermore, there was no initial evaluation on whether there is a difference between TCZ and BRT with respect to the risk of developing infections. In our study, we found no difference between the incidence of secondary infection in the TCZ and BRT groups. In addition, no risk factors were identified in the univariate analysis, which could be associated with the occurrence of secondary infection. Although we have not verified whether the complications of infections affected patient prognosis, neither TCZ nor BRT appear to pose a significant risk of infection.

Given the fact that treatment with either of the two drugs did not result in significantly different outcomes for patients, this may indicate that both drugs can be chosen with a comparable treatment in clinical practice. Baricitinib is an oral drug that can be administered even if the intravenous route is difficult to secure, and it is easy to discontinue. Tocilizumab is an intravenous or subcutaneous drug that can be used by patients with oral intake difficulty and those with severe renal dysfunction. The choice should be based on the characteristics of each drug in each individual patient.

Our study confirmed that the improvement in COVID-19 patients’ respiratory status following the administration of TCZ or BRT was similar. In contrast, we found that the use of anti-viral drugs was significantly associated with an improvement in respiratory status within 28 days from treatment initiation.

According to clinical trials, remdesivir has been observed to shorten the recovery duration for patients hospitalized with COVID-19 and with evidence of pneumonia [19]. In contrast, several studies have failed to show clear efficacy [20,21,22]; the efficacy of remdesivir as a single agent or adjunctive drug for standard care may be limited. Based on the results of our study, remdesivir may be an important drug that should essentially be administered to patients receiving biologic agents and steroids. This finding suggests the additive effect of remdesivir, which requires further investigation.

A relatively short duration from the onset of illness to the administration of the drugs was an independent factor for a poor prognosis. For both TCZ and BRT, relatively early administration of drugs in the trials was associated with relatively considerable reduction in the risk of death [7,11]. This apparently paradoxical observation was probably because the short time between the onset of symptoms and the administration of TCZ or BRT may reflect the rapid deterioration in respiratory status. In our hospital, biologic drugs are mostly administered to patients with increased oxygen demand and increased severity of illness. The deterioration of respiratory status early in the course of the disease may be a prognostic factor that cancels out the benefit of early administration of immunomodulators. The prognosis of patients who deteriorate rapidly after the onset of illness should be evaluated in future studies.

This study has several limitations. First, the study was retrospective. In consideration of confounding factors, we did not directly compare and verify the mortality or respiratory status in the groups of patients who used TCZ and BRT. Prospective validation is needed to show the comparative efficacy of the two drugs. Second, variant strains of SARS-CoV-2 and changes in healthcare availability that may affect patient outcomes were not validated in this study owing to the lack of data. Lastly, the efficacy of using TCZ and BRT as standalone treatments for COVID-19 was not verified in this study. However, the efficacy of TCZ and BRT has been proven in previous studies. Our study was conducted to suggest a more optimized treatment based on the aforementioned evidence-based practices of using TCZ and BRT.

## 5. Conclusions

In conclusion, neither tocilizumab nor baricitinib were associated with a higher risk of death or a greater contribution to an improved respiratory status. Prospective clinical trials need to be conducted to determine the superiority of both drugs.

## Figures and Tables

**Figure 1 medicina-58-00513-f001:**
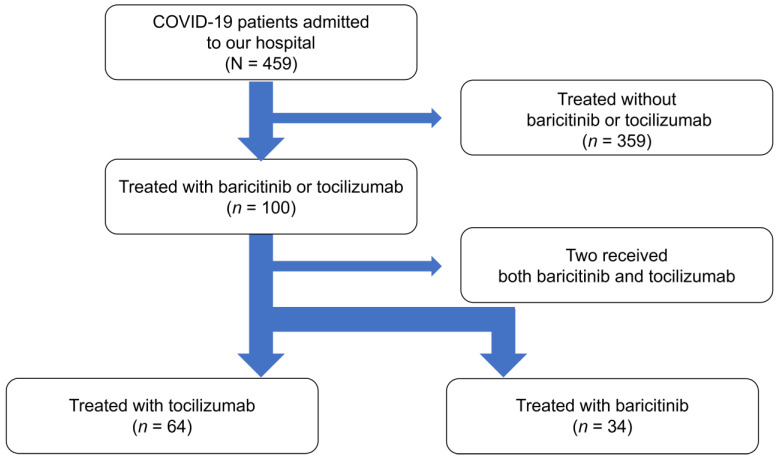
Flow chart of patients with coronavirus disease 2019 (COVID-19) in the tocilizumab and baricitinib groups. Coronavirus disease 2019, COVID-19.

**Figure 2 medicina-58-00513-f002:**
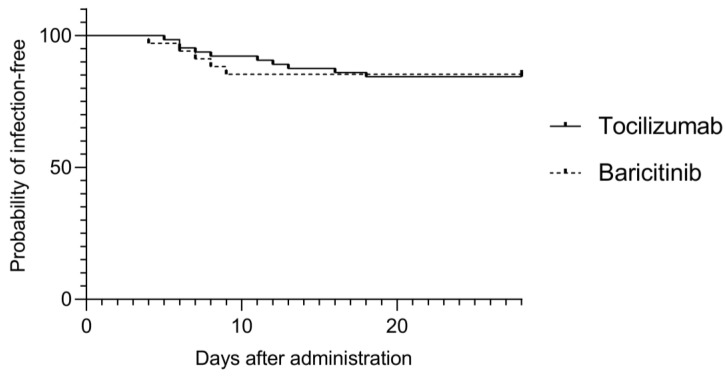
Kaplan–Meier estimates of secondary infection free survival between tocilizumab and baricitinib groups.

**Table 1 medicina-58-00513-t001:** Comparison between baseline characteristics of tocilizumab and baricitinib groups.

	Total (*n* = 98)	Tocilizumab (*n* = 64)	Baricitinib (*n* = 34)	*p*-Value
Characteristics				
Age (years)	60.5 (54.0, 70.3)	65.5 (54.3, 72.8)	58.5 (53.8, 64.3)	0.03
Sex (male)	73 (74.5)	46 (71.9)	27 (79.4)	0.42
Current smoker	16 (16.3)	9 (16.5)	7 (20.6)	0.41
BMI ≥ 30 (kg/m^2^) *	25 (27.2)	18 (30.0)	7 (20.6)	0.40
Chronic heart disease	17 (17.4)	15 (23.4)	2 (5.9)	0.03
Chronic kidney disease	6 (6.1)	4 (6.3)	2 (5.9)	0.94
Diabetes mellitus	36 (36.7)	23 (35.9)	13 (38.2)	0.82
Any collagen disease	3 (3.1)	3 (4.7)	0 (0)	0.20
Hypertension	46 (46.9)	34 (53.1)	12 (35.3)	0.09
Any respiratory disease	8 (8.1)	6 (9.4)	2 (5.9)	0.55
Immunosuppressive Drug regular use	3 (3.1)	2(3.1)	1 (2.9)	0.96
Fully vaccinated	1 (1.0)	0 (0)	1 (1.0)	0.14
Time from symptom onset to administration	9 (7, 12)	10 (7, 13)	9 (7, 11)	0.50
Time from onset to administration ≤ 7 days	29 (29.6)	19 (29.7)	10 (29.4)	0.98
Treatment				
Steroid	96 (98.0)	63 (98.4)	33 (97.1)	0.65
Heparin	78 (79.6)	55 (86.0)	23 (67.7)	0.03
Any anti-viral drug	75 (76.5)	45 (70.3)	30 (88.2)	0.046
Antibody cocktail therapy	1 (1.0)	0 (0)	1 (1.0)	0.14
Blood test at administration				
White blood cell (/μL)	7900 (5550, 10825)	8000 (5400, 11075)	7700 (5600, 10525)	0.64
Neutrophils (/μL) **	6991 (4465, 9755)	7173 (4716, 10027)	6780 (4418, 9280)	0.53
Lymphocytes (/μL) **	611 (476, 908)	648 (440, 911)	608 (528, 852)	0.65
Eosinophil (/μL) **	0 (0, 0)	0 (0, 0)	0 (0, 8.3)	0.047
Hemoglobin (g/dL)	14.2 (12.9, 15.0)	13.8 (12.8, 15.0)	14.5 (13.9, 15.6)	0.04
Platelet (×10^4^/μL)	19.0 (13.4, 25.9)	18.5 (12.7, 25.8)	20.7 (14.8, 26.4)	0.51
LDH (U/L)	512.5 (419.8, 647.5)	540 (438.8, 716.0)	470.5 (386.5, 603.5)	0.02
CRP (mg/mL)	7.1 (3.6, 11.2)	7.8 (3.9, 12.1)	5.8 (3.1, 9.4)	0.12
KL-6 (U/mL) ***	402 (289.5, 617.5)	444 (337, 705)	319 (240.3, 481.5)	0.03
Procalcitonin (ng/mL) ****	0.08 (0.05, 0.15)	0.08 (0.06, 0.12)	0.08 (0.05, 0.18)	0.97
Ferritin (ng/mL) *****	1125.5 (693.8, 1924.5)	1242.5 (745.3, 1966.3)	1080 (631.8, 1901)	0.44
D-dimer (μg/mL) ******	1.4 (1.0, 2.5)	1.5 (1.3, 3.1)	1.0 (0.8, 1.4)	<0.001
Severity				
1	1 (1)	1 (1.6)	0 (0)	
2	9 (9.2)	4 (6.3)	5 (14.7)	
3	76 (77.6)	49 (76.6)	27 (79.4)	
4	12 (12.2)	10 (15.6)	2 (5.9)	0.26
Outcomes				
Death within 28 days	14 (14.3)	13 (20.3)	1 (2.94)	
Improvement in respiratory status within 28 days	73 (74.5)	43 (67.2)	33 (88.2)	
Development of secondary infections	15 (15.3)	10 (15.6)	5 (14.7)	

* *n* = 92, ** *n* = 94, *** *n* = 69, **** *n* = 60, ***** *n* = 70, ****** *n* = 95; Data are shown as median (interquartile range) or number (%). We categorized COVID-19 severity at treatment initiation as follows; severity level 1: hospitalized but not requiring supplemental oxygen; severity level 2: hospitalized and requiring supplemental oxygen ≤4 L/min; severity level 3: hospitalized and requiring oxygen therapy ≥5 L/min or receiving nasal high-flow oxygen therapy, non-rebreather, or noninvasive mechanical ventilation; and severity level 4, receiving invasive mechanical ventilation.

**Table 2 medicina-58-00513-t002:** Mortality predicting factors among patients within 28 days from treatment with tocilizumab or baricitinib.

	Univariate		Multivariate	
	OR (95% CI)	*p*-Value	OR (95% CI)	*p*-Value
Characteristic				
Tocilizumab use *	8.41 (1.05–67.37)	0.045	13.28 (0.45–392.92)	0.13
Age (year)	1.10 (1.03–1.18)	0.004	1.10 (1.00–1.21)	0.04
Sex (male)	0.38 (0.12–1.26)	0.12		
Current smoker	0.35 (0.04–2.92)	0.33		
BMI ≥ 30 (kg/m^2^)	0.94 (0.27–3.30)	0.93		
Chronic heart disease	2.18 (0.59–8.03)	0.24		
Chronic kidney disease	16.40 (2.66–101.21)	0.003	43.10 (2.71–686.04)	0.008
Diabetes mellitus	1.35 (0.43–4.26)	0.61		
Any collagen disease	3.15 (0.27–37.31)	0.36		
Hypertension	2.29 (0.70–7.40)	0.17		
Any respiratory disease	4.31 (0.90–20.59)	0.07	1.85 (0.22–15.76)	0.57
Immunosuppressive drug regular use	3.15 (0.27–37.31)	0.36		
Time from onset to administration ≤ 7 days	5.76 (1.73–19.18)	0.004	18.09 (1.70–192.47)	0.02
Treatment				
Heparin	1.64 (0.34–7.98)	0.54		
Any anti-viral drug	0.34 (0.10–1.11)	0.07	0.16 (0.01–1.96)	0.15
Blood test at administration				
Lymphocytes (×10^3^/μL)	0.36 (0.01–13.81)	0.58		
Platelet (×10^5^/μL)	0.66 (0.35–1.24)	0.20		
LDH (×10^2^ U/L)	1.18 (0.88–1.59)	0.27		
CRP (mg/mL)	1.02 (0.92–1.13)	0.68		
KL–6 (×10^2^ U/mL)	1.06 (0.87–1.30)	0.57		
Procalcitonin (ng/mL)	1.36 (0.18–9.97)	0.76		
Ferritin (×10^3^ ng/mL)	0.37 (0.10–1.47)	0.16		
D-dimer (μg/mL)	0.10 (0.96–1.03)	0.82		
Severity				
1 or 2	(reference)	(reference)		
3	1.52 (0.18–13.24)	0.70		
4	1.80 (0.14–23.37)	0.65		

* Analysis among 98 patients using either tocilizumab or baricitinib. We categorized COVID-19 severity at treatment initiation as follows, severity level 1: hospitalized but not requiring supplemental oxygen; severity level 2: hospitalized and requiring supplemental oxygen ≤4 L/min; severity level 3: hospitalized and requiring oxygen therapy ≥5 L/min or receiving nasal high-flow oxygen therapy, non-rebreather, or noninvasive mechanical ventilation; and severity level 4: receiving invasive mechanical. Logistic regression analysis was used. BMI, body mass index; CI, Confidence interval; CRP, C-reactive protein; KL-6, Krebs von den Lungen-6; LDH, lactate dehydrogenase; OR, odds ratio.

**Table 3 medicina-58-00513-t003:** Predicting factors for respiratory improvement within 28 days from administration of tocilizumab or baricitinib treatment.

	Univariate		Multivariate	
	OR (95% CI)	*p*-Value	OR (95% CI)	*p*-Value
Characteristics				
Baricitinib use *	3.66 (1.14–11.76)	0.03	1.75 (0.35–8.67)	0.50
Age (year)	0.94 (0.89–0.98)	0.004	0.94 (0.88–1.01)	0.07
Sex (male)	1.55 (0.57–4.22)	0.39		
Current smoker	6.21 (0.78–49.65)	0.09	4.14 (0.28–60.37)	0.29
BMI ≥ 30 (kg/m^2^)	1.46 (0.47–4.49)	0.50		
Chronic heart disease	0.27 (0.09–0.80)	0.03	0.40 (0.09–1.89)	0.25
Chronic kidney disease	0.15 (0.03–0.86)	0.03	0.12 (0.01–1.80)	0.13
Diabetes mellitus	1.04 (0.40–2.68)	0.93		
Any collagen disease	0.68 (0.07–9.12)	0.75		
Hypertension	0.39 (0.15–1.01)	0.047	0.84 (0.20–3.59)	0.82
Any respiratory disease	0.54 (0.12–2.44)	0.42		
Immunosuppressive drug regular use	0.67 (0.05–7.79)	0.75		
Time from onset to administration ≤ 7 days	0.26 (0.10–0.68)	0.006	0.82 (0.02–0.40)	0.002
Treatment				
Heparin	0.97 (0.31–3.00)	0.95		
Any anti-viral drug	3.99 (1.46–10.89)	0.007	6.5 (1.13–37.56)	0.04
Blood test at administration				
Lymphocytes (×10^3^/μL)	1.07 (0.92–1.24)	0.37		
Platelet (×10^5^/μL)	1.37 (0.84–2.24)	0.20		
LDH (×10^2^ U/L)	0.85 (0.66–1.09)	0.19		
CRP (mg/mL)	1.01 (0.92–1.09)	0.91		
KL–6 (×10^2^ U/mL)	0.91 (0.78–1.06)	0.25		
Procalcitonin (ng/mL)	1.08 (0.13–12.74)	0.83		
Ferritin (ng/mL)	1.15 (0.65–2.03)	0.64		
D–dimer (μg/mL)	0.98 (0.95–1.00)	0.09	0.98 (0.96–1.01)	0.25
Severity				
1 or 2	(reference)	(reference)	(reference)	(reference)
3	0.36 (0.04–3.02)	0.35		
4	0.11 (0.01–1.17)	0.07	0.04 (0.002–1.01)	0.05

* Analysis among 98 patients using either tocilizumab or baricitinib. We defined COVID-19 severity at treatment initiation as follows: severity level 1: hospitalized but not requiring supplemental oxygen; severity level 2: hospitalized and requiring supplemental oxygen ≤ 4 L/min; severity level 3: hospitalized and requiring oxygen therapy ≥ 5 L/min or receiving nasal high-flow oxygen therapy, non-rebreather, or noninvasive mechanical ventilation; and severity level 4: receiving invasive mechanical ventilation. Logistic regression analysis was used. BMI, body mass index; CI, Confidence interval; CRP, C-reactive protein; KL-6, Krebs von den Lungen-6; LDH, lactate dehydrogenase; OR, odds ratio.

## Data Availability

The data presented in this study are available on request from the corresponding author.

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
