# Peer review of "Comparative Efficacy of Tocilizumab and Baricitinib Administration in COVID-19 Treatment: A Retrospective Cohort Study"

_medicina, 2022, doi:10.3390/medicina58040513_

Round 1

Reviewer 1 Report

Kojima and colleagues' retrospective work examined the usefulness of the drugs tocilizumab and baricitinib during Covid-19 infection with the question of the extent to which taking the drugs can lead to respiratory improvement. They studied these administered drugs in a cohort of 459 patients and ultimately concluded that neither drug had a good effect on disease severity or mortality. There are already a total of many drug studies in this episode that are working out this issue to reduce disease severity. For example, in the past, the use of so-called antimalarials or other therapeutics was given much importance. Nonetheless, the work itself is exciting because it shows how complex the course of covid-19 patients can have and how these two drugs may not work. I would suggest more focus on the molecular aspects of the drugs in the introduction, again compare current drugs routinely used in critical care. A small meta-analysis could be beneficial here, which could also serve as a basis for discussion, as some of the discussion is redundant and not precisely related to the results. The conclusion should be presented in detail and should also include a detailed limitation section summarizing the strengths and limitations of the study.

Author Response

Response to Reviewer

Reviewer 1:

Kojima and colleagues' retrospective work examined the usefulness of the drugs tocilizumab and baricitinib during Covid-19 infection with the question of the extent to which taking the drugs can lead to respiratory improvement. They studied these administered drugs in a cohort of 459 patients and ultimately concluded that neither drug had a good effect on disease severity or mortality. There are already a total of many drug studies in this episode that are working out this issue to reduce disease severity. For example, in the past, the use of so-called antimalarials or other therapeutics was given much importance. Nonetheless, the work itself is exciting because it shows how complex the course of covid-19 patients can have and how these two drugs may not work.

Reply: Thank you for reviewing our manuscript.

We would like to explain the background of this study. Our practice policy for COVID-19 is based on the results of RCTs and recommendations of various guidelines, and since tocilizumab and baricitinib have been shown to be effective in severe COVID-19 cases, we have also used these agents. However, although the clinical efficacy of both agents has been shown in independent RCTs, it remains unclear which agent is more effective when compared directly. In clinical practice, we often wondered whether tocilizumab or baricitinib should be used. This study was designed to answer this clinical question.

The conclusions of this study are not that either tocilizumab or baricitinib is ineffective. The study was limited to a group of patients treated with either tocilizumab or baricitinib, and there was no evidence of a higher risk of death or a lower contribution to an improved respiratory status for either drug.

Your point made us realize that the results of this study could be interpreted as "the mortality risk of the drug itself" or "the contribution of the drug itself to improving respiratory status".

We have added a description to the Abstract, Results and Discussion to clarify the results and intent of this study.  (Line 29, 124-125, 160, 175-176, 188-190, 191-193, 196-197, 208)

I would suggest more focus on the molecular aspects of the drugs in the introduction, again compare current drugs routinely used in critical care.

Reply: We have added information about the mechanism of various immunomodulators, including steroids, as suggested. (Line 45-46, 51-52, 57-59)

A small meta-analysis could be beneficial here, which could also serve as a basis for discussion, as some of the discussion is redundant and not precisely related to the results.

Reply: In the Discussion, we interpreted the results obtained based on those of previous reports; please excuse the length of the text as we have identified a variety of relevant factors as well as the impact of TCZ and BRT on the outcomes.

The conclusion should be presented in detail and should also include a detailed limitation section summarizing the strengths and limitations of the study.

Reply: The Conclusions have been changed to objective statements based on the results of the study. In the Limitations, we describe in more detail the limitations of the retrospective nature of our study. (Line 264-267, 275-277)

Reviewer 2 Report

This study aims to compare the efficacy of tocilizumab and baricitinib in the treatment of COVID-19. However, the methodology adopted cannot answer the research questions. The conclusions are also not supported by the data presented.

Major comments:

1. The study aims to compare the efficacy of tocilizumab and baricitinib. However, it does not provide any direct comparison of the efficacy of the two drugs (e.g., comparison of the mortality rate or the respiratory improvement rate). It is unclear why the authors tried to identify risk factors for mortality and respiratory improvement, as this is not relevant to the research question. The efficacy of each drug has already been reported in various RCTs.

2. It is even more confusing why the authors did not include baricitinib use in the analysis of mortality, and tocilizumab use in respiratory improvement.

3. Some key data were missing, including number of the primary outcome events (death, respiratory improvement) and information about treatment administration (time of initiation, dosage, duration).

4. The authors conclude “there were no differences between the efficacy and safety levels of tocilizumab and baricitinib for COVID-19 treatment”. However, there were no data to support such conclusions.

Minor comments:

The term “biological agents” may also refer to anti-SARS-CoV-2 monoclonal antibodies or convalescent plasma. Also, baricitinib is not a biological agent.

The word “infection-free” is ambiguous. Please indicate clearly that this refers to secondary infections.

Line 66-68: This is the study conclusion, which should not be in the introduction session.

Line 99: Does it mean “recovery to severity levels 1 or 2”? How to determine respiratory improvement in patients with baseline severity levels of 1 or 2?

Author Response

Response to Reviewer

Reviewer 2:

Comments and Suggestions for Authors

This study aims to compare the efficacy of tocilizumab and baricitinib in the treatment of COVID-19. However, the methodology adopted cannot answer the research questions. The conclusions are also not supported by the data presented.

Major comments:

  1. The study aims to compare the efficacy of tocilizumab and baricitinib. However, it does not provide any direct comparison of the efficacy of the two drugs (e.g., comparison of the mortality rate or the respiratory improvement rate). It is unclear why the authors tried to identify risk factors for mortality and respiratory improvement, as this is not relevant to the research question. The efficacy of each drug has already been reported in various RCTs.
  2. It is even more confusing why the authors did not include baricitinib use in the analysis of mortality, and tocilizumab use in respiratory improvement.

Reply: Thank you for reviewing our manuscript. As pointed out, the efficacy of tocilizumab and baricitinib has been demonstrated in previous RCTs, and this was noted in the Introduction.

In clinical practice, however, the choice between the two drugs is often a difficult one. Although both are recommended, there is no indication on when to use which of the two drugs. To answer this clinical question, we analyzed the limited data that were available, using clinical information from COVID-19 patients treated at our hospital.

First of all, we would like clarify that 98 patients who exclusively used tocilizumab or baricitinib were included in the analysis (64 in the tocilizumab group and 34 in the baricitinib group). Therefore, all patients who did not fall into the "tocilizumab use" category were patients who fell into the "baricitinib use" category. The risk for the "tocilizumab use" outcome in the logistic regression analysis is the risk compared to that of "baricitinib use", i.e., both drugs were compared directly.

Comparing the mortality and improvements in respiratory status of the groups using each drug, the tocilizumab group in this study had a clearly higher mortality rate. However, the decision to use the drugs was not made randomly in clinical practice, and there was a bias introduced by patient background. In COVID-19, age and underlying diseases have a significant impact on mortality. As this was a retrospective study, logistic regression analysis was performed with the aim of eliminating the influence of these confounding factors.

However, we realized that the statements in the Abstract, Results and Discussion of this study were confusing at times. We have revised and added text to make it easier for the reader to understand the results and intentions we present. (Line 29, 124-125, 160, 175-176, 188-190, 191-193, 196-197, 208)

The captions of Tables 2 and 3 were amended to state that analysis was performed on 98 patients who were treated with either tocilizumab or baricitinib. (Line 377, 392)

  1. Some key data were missing, including number of the primary outcome events (death, respiratory improvement) and information about treatment administration (time of initiation, dosage, duration).

Reply: In response to the suggestion, we have added data on mortality and improvement in respiratory status to the Results and Table 1. The dosage and duration for each drug, which were consistent for almost all patients, were described in the Results. (Line 153-156, 140-145, Table 1)

  1. The authors conclude “there were no differences between the efficacy and safety levels of tocilizumab and baricitinib for COVID-19 treatment”. However, there were no data to support such conclusions.

Reply: As pointed out, given that the efficacy of the respective drug itself for COVID-19 treatment has not been verified, our statement was inappropriate. It was also misleading to use the expression "safety" based solely on the presence or absence of infectious disease events. We revised the description. (Line 35-36, 275-277)

Minor comments:

The term “biological agents” may also refer to anti-SARS-CoV-2 monoclonal antibodies or convalescent plasma. Also, baricitinib is not a biological agent.

Reply: As pointed out, we should have avoided referring to baricitinib, a JAK inhibitor, as a biological agent. We have used a different term for all references to "biological agents" in the text. We considered using the term “immunomodulator”, but since it includes steroids, we have only used it in parts of the revised text.

The word “infection-free” is ambiguous. Please indicate clearly that this refers to secondary infections.

Reply: The Figure caption was ambiguous, as pointed out, so that we have revised the term to "secondary infection". (Line 408-409)

Line 66-68: This is the study conclusion, which should not be in the introduction session.

Reply: We have removed it from the relevant section, as suggested.

Line 99: Does it mean “recovery to severity levels 1 or 2”? How to determine respiratory improvement in patients with baseline severity levels of 1 or 2?

Reply: In this study, patients with an initial respiratory status of severity 1 or 2 were considered "improved" if the status was still 1 or 2 at day 28. We have added the respective information to the Methods. (Line 102-103)

Round 2

Reviewer 1 Report

The authors have well addressed all my concerns. The impact of this manuscript has been sig nificantly increased. Good work!

Author Response

You have helped us to make a better research paper. Thank you very much.

Reviewer 2 Report

Thanks for the thoughtful reply. The authors have addressed the previous questions and made proper revisions to the article. I still have the following concerns:

1. There were 6 variables included in the multivariate logistic model. However, only 14 death events were observed. Usually, it requires at least 5 events per variable. This is even more non-reassuring given that the univariate and multivariate analyses yielded widely different results. The authors may want to limit the number of covariates, or use other statistic methods such as propensity score weighting to control for confounding factors.

2. The authors stated “Age, chronic renal disease, and comorbid respiratory disease were independent prognostic factors for all-cause mortality”. However, respiratory disease was not associated with mortality in the multivariable model.

3. The authors did not mention the results of “Time from onset to administration ≤ 7 days” in the discussion and conclusions, which was significant in both models.

Author Response

  1. There were 6 variables included in the multivariate logistic model. However, only 14 death events were observed. Usually, it requires at least 5 events per variable. This is even more non-reassuring given that the univariate and multivariate analyses yielded widely different results. The authors may want to limit the number of covariates, or use other statistic methods such as propensity score weighting to control for confounding factors.

Thank you once again for reviewing our manuscript.

As you noted, the number of cases in this study is limited and therefore the number of events of death is also small. On the other hand, we were hesitant to exclude from the analysis some factors that have been clearly shown to be associated with death (age, underlying disease, baseline severity, etc.). Therefore, we followed your recommendation and performed sensitivity analysis using the propensity score, resulting in 25 matched cases in each group. The mortality, respiratory status recovery rates, and secondary infection rate were compared, and no significant differences were found.

This result has been added to the text and shown as supplementary materials (Line 117-119, 191-199, Table S2).

  1. The authors stated “Age, chronic renal disease, and comorbid respiratory disease were independent prognostic factors for all-cause mortality”. However, respiratory disease was not associated with mortality in the multivariable model.

In the Abstract we listed "respiratory disease" as a prognostic factor, which we realized was incorrect. We have corrected it. We appreciate your pointing this out. (Line 31-32)

  1. The authors did not mention the results of “Time from onset to administration ≤ 7 days” in the discussion and conclusions, which was significant in both models.

Regarding your point, we have included this in Discussion since the initial draft. At first glance, the results seem to indicate that the prognosis is worse for the group receiving earlier treatment. However, the timing of initiation of tocilizumab or baricitinib roughly coincides with the time of worsening respiratory status. The group that started treatment earlier can be interpreted as having a shorter time to worsening respiratory status, i.e., a group that deteriorated more rapidly. This is discussed in the Discussion (Line 268-278).